# Learning Multimodal Graph-to-Graph Translation for Molecular Optimization

**Wengong Jin, Kevin Yang, Regina Barzilay, Tommi Jaakkola**
Computer Science and Artificial Intelligence Lab, Massachusetts Institute of Technology
{wengong, regina, tommi}@csail.mit.edu;  yangk@mit.edu

## Abstract

We view molecular optimization as a graph-to-graph translation problem. The goal is to learn to map from one molecular graph to another with better properties based on an available corpus of paired molecules. Since molecules can be optimized in different ways, there are multiple viable translations for each input graph. A key challenge is therefore to model diverse translation outputs. Our primary contributions include a junction tree encoder-decoder for learning diverse graph translations along with a novel adversarial training method for aligning distributions of molecules. Diverse output distributions in our model are explicitly realized by low-dimensional latent vectors that modulate the translation process. We evaluate our model on multiple molecular optimization tasks and show that our model outperforms previous state-of-the-art baselines.

## 1 Introduction

The goal of drug discovery is to design molecules with desirable chemical properties. The task is challenging since the chemical space is vast and often difficult to navigate. One of the prevailing approaches, known as *matched molecular pair analysis* (MMPA) (Griffen et al., 2011; Dossetter et al., 2013), learns rules for generating "molecular paraphrases" that are likely to improve target chemical properties. The setup is analogous to machine translation: MMPA takes as input molecular pairs $\{(X, Y)\}$, where $Y$ is a paraphrase of $X$ with better chemical properties. However, current MMPA methods distill the matched pairs into graph transformation rules rather than treating it as a general translation problem over graphs based on parallel data.

In this paper, we formulate molecular optimization as *graph-to-graph translation*. Given a corpus of molecular pairs, our goal is to learn to translate input molecular graphs into better graphs. The proposed translation task involves many challenges. While several methods are available to encode graphs (Duvenaud et al., 2015; Li et al., 2015; Lei et al., 2017), generating graphs as output is more challenging without resorting to a domain-specific graph linearization. In addition, the target molecular paraphrases are diverse since multiple strategies can be applied to improve a molecule. Therefore, our goal is to learn multimodal output distributions over graphs.

To this end, we propose *junction tree encoder-decoder*, a refined graph-to-graph neural architecture that decodes molecular graphs with neural attention. To capture diverse outputs, we introduce stochastic latent codes into the decoding process and guide these codes to capture meaningful molecular variations. The basic learning problem can be cast as a variational autoencoder, where the posterior over the latent codes is inferred from input molecular pair $(X, Y)$. Further, to avoid invalid translations, we propose a novel adversarial training method to align the distribution of graphs generated from the model using randomly selected latent codes with the observed distribution of valid targets. Specifically, we perform adversarial regularization on the level of the hidden states created as part of the graph generation.

We evaluate our model on three molecular optimization tasks, with target properties ranging from drug likeness to biological activity.[1] As baselines, we utilize state-of-the-art graph generation methods (Jin et al., 2018; You et al., 2018a) and MMPA (Dalke et al., 2018). We demonstrate that our model excels in discovering molecules with desired properties, outperforming the baselines across

---

[1]Code and data are available at `https://github.com/wengong-jin/iclr19-graph2graph`

different tasks. Meanwhile, our model can translate a given molecule into a diverse set of compounds, demonstrating the diversity of learned output distributions.

## 2 RELATED WORK

**Molecular Generation/Optimization** Prior work on molecular optimization approached the graph translation task through generative modeling (Gómez-Bombarelli et al., 2016; Segler et al., 2017; Kusner et al., 2017; Dai et al., 2018; Jin et al., 2018; Samanta et al., 2018; Li et al., 2018a) and reinforcement learning (Guimaraes et al., 2017; Olivecrona et al., 2017; Popova et al., 2018; You et al., 2018a). Earlier approaches represented molecules as SMILES strings (Weininger, 1988), while more recent methods represented them as graphs. Most of these methods coupled a molecule generator with a property predictor and solved the optimization problem through Bayesian optimization or reinforcement learning. In contrast, our model is trained to translate a molecular graph into a better graph through supervised learning, which is more sample efficient.

Our approach is closely related to matched molecular pair analysis (MMPA) (Griffen et al., 2011; Dossetter et al., 2013) in drug de novo design, where the matched pairs are hard-coded into graph transformation rules. MMPA's main drawback is that large numbers of rules have to be realized (e.g. millions) to cover all the complex transformation patterns. In contrast, our approach uses neural networks to learn such transformations, which does not require the rules to be explicitly realized.

**Graph Neural Networks** Our work is related to graph encoders and decoders. Previous work on graph encoders includes convolutional (Scarselli et al., 2009; Bruna et al., 2013; Henaff et al., 2015; Duvenaud et al., 2015; Niepert et al., 2016; Defferrard et al., 2016; Kondor et al., 2018) and recurrent architectures (Li et al., 2015; Dai et al., 2016; Lei et al., 2017). Graph encoders have been applied to social network analysis (Kipf & Welling, 2016; Hamilton et al., 2017) and chemistry (Kearnes et al., 2016; Gilmer et al., 2017; Schütt et al., 2017; Jin et al., 2017). Recently proposed graph decoders (Simonovsky & Komodakis, 2018; Li et al., 2018b; Jin et al., 2018; You et al., 2018b; Liu et al., 2018) focus on learning generative models of graphs. While our model builds on Jin et al. (2018) to generate graphs, we contribute new techniques to learn multimodal graph-to-graph mappings.

**Image/Text Style Translation** Our work is closely related to image-to-image translation (Isola et al., 2017), which was later extended by Zhu et al. (2017) to learn multimodal mappings. Our adversarial training technique is inspired by recent text style transfer methods (Shen et al., 2017; Zhao et al., 2018) that adversarially regularize the continuous representation of discrete structures to enable end-to-end training. Our technical contribution is a novel adversarial regularization over graphs that constrains their scaffold structures in a continuous manner.

## 3 JUNCTION TREE ENCODER-DECODER

Our translation model extends the junction tree variational autoencoder (Jin et al., 2018) to an encoder-decoder architecture for learning graph-to-graph mappings. Following their work, we interpret each molecule as having been built from subgraphs (clusters of atoms) chosen from a vocabulary of valid chemical substructures. The clusters form a junction tree representing the *scaffold* structure of molecules (Figure 1), which is an important factor in drug design. Molecules are decoded hierarchically by first generating the junction trees and then combining the nodes of the tree into a molecule. This coarse-to-fine approach allows us to easily enforce the chemical validity of generated graphs, and provides an enriched representation that encodes molecules at different scales.

In terms of model architecture, the encoder is a graph message passing network that embeds both nodes in the tree and graph into continuous vectors. The decoder consists of a tree-structured decoder for predicting junction trees, and a graph decoder that learns to combine clusters in the predicted junction tree into a molecule. Our key departures from Jin et al. (2018) include a unified encoder architecture for trees and graphs, along with an attention mechanism in the tree decoding process.

### 3.1 TREE AND GRAPH ENCODER

Viewing trees as graphs, we encode both junction trees and graphs using graph message passing networks. Specifically, a graph is defined as $G = (\mathcal{V}, \mathcal{E})$ where $\mathcal{V}$ is the vertex set and $\mathcal{E}$ the edge

Figure 1: Illustration of our encoder-decoder model. Molecules are represented by their graph structures and junction trees encoding the *scaffold* of molecules. Nodes in the junction tree (which we call *clusters*) are valid chemical substructures such as rings and bonds. During decoding, the model first generates its junction tree and then combines clusters in the predicted tree into a molecule.

set. Each node $v$ has a feature vector $\boldsymbol{f}_v$. For atoms, it includes the atom type, valence, and other atomic properties. For clusters in the junction tree, $\boldsymbol{f}_v$ is a one-hot vector indicating its cluster label. Similarly, each edge $(u, v) \in \mathcal{E}$ has a feature vector $\boldsymbol{f}_{uv}$. Let $N(v)$ be the set of neighbor nodes of $v$. There are two hidden vectors $\boldsymbol{\nu}_{uv}$ and $\boldsymbol{\nu}_{vu}$ for each edge $(u, v)$ representing the message from $u$ to $v$ and vice versa. These messages are updated iteratively via neural network $g_1(\cdot)$:

$$\boldsymbol{\nu}_{uv}^{(t)} = g_1\left(\boldsymbol{f}_u, \boldsymbol{f}_{uv}, \sum\nolimits_{w \in N(u) \backslash v} \boldsymbol{\nu}_{wu}^{(t-1)}\right) \tag{1}$$

where $\boldsymbol{\nu}_{uv}^{(t)}$ is the message computed in the $t$-th iteration, initialized with $\boldsymbol{\nu}_{uv}^{(0)} = \mathbf{0}$. In each iteration, all messages are updated asynchronously, as there is no natural order among the nodes. This is different from the tree encoding algorithm in Jin et al. (2018), where a root node was specified and an artificial order was imposed on the message updates. Removing this artifact is necessary as the learned embeddings will be biased by the artificial order.

After $T$ steps of iteration, we aggregate messages via another neural network $g_2(\cdot)$ to derive the latent vector of each vertex, which captures its local graph (or tree) structure:

$$\boldsymbol{x}_u = g_2\left(\boldsymbol{f}_u, \sum\nolimits_{v \in N(u)} \boldsymbol{\nu}_{vu}^{(T)}\right) \tag{2}$$

Applying the above message passing network to junction tree $\mathcal{T}$ and graph $G$ yields two sets of vectors $\{\boldsymbol{x}_1^{\mathcal{T}}, \cdots, \boldsymbol{x}_n^{\mathcal{T}}\}$ and $\{\boldsymbol{x}_1^{\mathcal{G}}, \cdots, \boldsymbol{x}_n^{\mathcal{G}}\}$. The *tree vector* $\boldsymbol{x}_i^{\mathcal{T}}$ is the embedding of tree node $i$, and the *graph vector* $\boldsymbol{x}_j^{\mathcal{G}}$ is the embedding of graph node $j$.

## 3.2 JUNCTION TREE DECODER

We generate a junction tree $\mathcal{T} = (\mathcal{V}, \mathcal{E})$ with a tree recurrent neural network with an attention mechanism. The tree is constructed in a top-down fashion by expanding the tree one node at a time. Formally, let $\tilde{\mathcal{E}} = \{(i_1, j_1), \cdots, (i_m, j_m)\}$ be the edges traversed in a depth first traversal over tree $\mathcal{T}$, where $m = 2|\mathcal{E}|$ as each edge is traversed in both directions. Let $\tilde{\mathcal{E}}_t$ be the first $t$ edges in $\tilde{\mathcal{E}}$. At the $t$-th decoding step, the model visits node $i_t$ and receives message vectors $\boldsymbol{h}_{ij}$ from its neighbors. The message $\boldsymbol{h}_{i_t, j_t}$ is updated through a tree Gated Recurrent Unit (Jin et al., 2018):

$$\boldsymbol{h}_{i_t, j_t} = \mathrm{GRU}(\boldsymbol{f}_{i_t}, \{\boldsymbol{h}_{k, i_t}\}_{(k, i_t) \in \tilde{\mathcal{E}}_t, k \neq j_t}) \tag{3}$$

**Topological Prediction** When the model visits node $i_t$, it first computes a predictive hidden state $\boldsymbol{h}_t$ by combining node features $\boldsymbol{f}_{i_t}$ and inward messages $\{\boldsymbol{h}_{k, i_t}\}$ via a one hidden layer network. The model then makes a binary prediction on whether to expand a new node or backtrack to the parent of $i_t$. This probability is computed by aggregating the source encodings $\{\boldsymbol{x}_*^{\mathcal{T}}\}$ and $\{\boldsymbol{x}_*^{\mathcal{G}}\}$ through an attention layer, followed by a feed-forward network ($\tau(\cdot)$ stands for ReLU and $\sigma(\cdot)$ for sigmoid):

$$\boldsymbol{h}_t = \tau(\boldsymbol{W}_1^d \boldsymbol{f}_{i_t} + \boldsymbol{W}_2^d \sum\nolimits_{(k, i_t) \in \tilde{\mathcal{E}}_t} \boldsymbol{h}_{k, i_t}) \tag{4}$$

$$\boldsymbol{c}_t^d = \mathrm{attention}\left(\boldsymbol{h}_t, \{\boldsymbol{x}_*^{\mathcal{T}}\}, \{\boldsymbol{x}_*^{\mathcal{G}}\}; \boldsymbol{U}_{att}^d\right) \tag{5}$$

$$\boldsymbol{p}_t = \sigma\left(\boldsymbol{u}^d \cdot \tau(\boldsymbol{W}_3^d \boldsymbol{h}_t + \boldsymbol{W}_4^d \boldsymbol{c}_t^d)\right) \tag{6}$$

Here we use $\mathrm{attention}(\cdot; \boldsymbol{U}_{att}^d)$ to mean the attention mechanism with parameters $\boldsymbol{U}_{att}^d$. It computes two set of attention scores $\{\boldsymbol{\alpha}_*^{\mathcal{T}}\}, \{\boldsymbol{\alpha}_*^{\mathcal{G}}\}$ (normalized by softmax) over source tree and graph vectors respectively. The output $\boldsymbol{c}_t^d$ is a concatenation of tree and graph attention vectors:

$$c_t^d = \left[ \sum_i \boldsymbol{\alpha}_{i,t}^{\mathcal{T}} \boldsymbol{x}_i^{\mathcal{T}}, \sum_i \boldsymbol{\alpha}_{i,t}^{\mathcal{G}} \boldsymbol{x}_i^{\mathcal{G}} \right] \qquad (7)$$

**Label Prediction** If node $j_t$ is a new child to be generated from parent $i_t$, we predict its label by

$$\boldsymbol{c}_t^l = \text{attention}(\boldsymbol{h}_{i_t,j_t}, \{\boldsymbol{x}_*^{\mathcal{T}}\}, \{\boldsymbol{x}_*^{\mathcal{G}}\}; \boldsymbol{U}_{att}^l) \quad (8)$$

$$\boldsymbol{q}_t = \text{softmax}(\boldsymbol{U}^l \tau(\boldsymbol{W}_1^l \boldsymbol{h}_{i_t,j_t} + \boldsymbol{W}_2^l \boldsymbol{c}_t^l)) \quad (9)$$

where $\boldsymbol{q}_t$ is a distribution over the label vocabulary and $\boldsymbol{U}_{att}^l$ is another set of attention parameters.

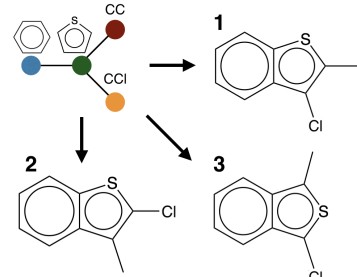

Figure 2: Multiple ways to assemble neighboring clusters in the junction tree.

### 3.3 GRAPH DECODER

The second step in the decoding process is to construct a molecular graph $G$ from a predicted junction tree $\widehat{\mathcal{T}}$. This step is not deterministic since multiple molecules could correspond to the same junction tree. For instance, the junction tree in Figure 2 can be assembled into three different molecules. The underlying degree of freedom pertains to how neighboring clusters are attached to each other. Let $\mathcal{G}_i$ be the set of possible candidate attachments at tree node $i$. Each graph $G_i \in \mathcal{G}_i$ is a particular realization of how cluster $C_i$ is attached to its neighboring clusters $\{C_j, j \in N_{\widehat{\mathcal{T}}}(i)\}$. The goal of the graph decoder is to predict the correct attachment between the clusters.

To this end, we design the following scoring function $f(\cdot)$ for ranking candidate attachments within the set $\mathcal{G}_i$. We first apply a graph message passing network over graph $G_i$ to compute atom representations $\{\boldsymbol{\mu}_v^{G_i}\}$. Then we derive a vector representation of $G_i$ through sum-pooling: $\boldsymbol{m}_{G_i} = \sum_v \boldsymbol{\mu}_v^{G_i}$. Finally, we score candidate $G_i$ by computing dot products between $\boldsymbol{m}_{G_i}$ and the encoded source graph vectors: $f(G_i) = \sum_{u \in G} \boldsymbol{m}_{G_i} \cdot \boldsymbol{x}_u^{\mathcal{G}}$.

The graph decoder is trained to maximize the log-likelihood of ground truth subgraphs at all tree nodes (Eq. (10)). During training, we apply teacher forcing by feeding the graph decoder with ground truth junction tree as input. During testing, we assemble the graph one neighborhood at a time, following the order in which the junction tree was decoded.

$$\mathcal{L}_g(G) = \sum_i \left[ f(G_i) - \log \sum_{G_i' \in \mathcal{G}_i} \exp(f(G_i')) \right] \qquad (10)$$

## 4 MULTIMODAL GRAPH-TO-GRAPH TRANSLATION

Our goal is to learn a multimodal mapping between two molecule domains, such as molecules with low and high solubility, or molecules that are potent and impotent. During training, we are given a dataset of paired molecules $\{(X, Y)\} \subset \mathcal{X} \times \mathcal{Y}$ sampled from their joint distribution $P(\mathcal{X}, \mathcal{Y})$, where $\mathcal{X}, \mathcal{Y}$ are the source and target domains. It is important to note that this joint distribution is a many-to-many mapping. For instance, there exist many ways to modify molecule $X$ to increase its solubility. Given a new molecule $X$, the model should be able to generate a diverse set of outputs.

To this end, we propose to augment the basic encoder-decoder model with low-dimensional latent vectors $\boldsymbol{z}$ to explicitly encode the multimodal aspect of the output distribution. The mapping to be learned now becomes $\mathcal{F} : (X, \boldsymbol{z}) \to Y$, with latent code $\boldsymbol{z}$ drawn from a prior distribution $P(\boldsymbol{z})$, which is a standard Gaussian $\mathcal{N}(\boldsymbol{0}, \boldsymbol{I})$. There are two challenges in learning this mapping. First, as shown in the image domain (Zhu et al., 2017), the latent codes are often ignored by the model unless we explicitly enforce the latent codes to encode meaningful variations. Second, the model should be properly regularized so that it does not produce invalid translations. That is, the translated molecule $\mathcal{F}(X, \boldsymbol{z})$ should always belong to the target domain $\mathcal{Y}$ given latent code $\boldsymbol{z} \sim \mathcal{N}(\boldsymbol{0}, \boldsymbol{I})$. In this section, we propose two techniques to address these issues.

### 4.1 VARIATIONAL JUNCTION TREE ENCODER-DECODER (VJTNN)

First, to encode meaningful variations, we derive latent code $\boldsymbol{z}$ from the embedding of ground truth molecule $Y$. The decoder is trained to reconstruct $Y$ when taking as input both its vector encoding

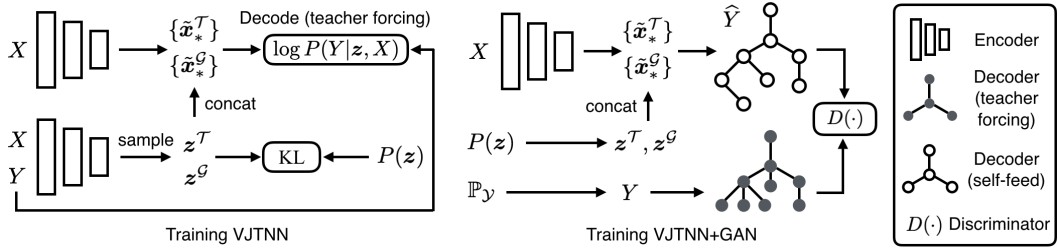

Figure 3: Multimodal graph-to-graph learning. Our model combines the strength of both variational JTNN and adversarial scaffold regularization.

$z_Y$ and source molecule $X$. For efficient sampling, the latent code distribution is regularized to be close to the prior distribution, similar to a variational autoencoder. We also restrict $z_Y$ to be a low dimensional vector to prevent the model from ignoring input $X$ and degenerating to an autoencoder.

Specifically, we first embed molecules $X$ and $Y$ into their tree and graph vectors $\{x_*^{\mathcal{T}}\}, \{x_*^{\mathcal{G}}\}$; $\{y_*^{\mathcal{T}}\}, \{y_*^{\mathcal{G}}\}$, using the same encoder with shared parameters (Sec 3.1). Then we compute the difference vector $\delta_{X,Y}$ between molecules $X$ and $Y$ as in Eq.(11). Since each tree and graph vector $y_i$ represents local substructure in the junction tree and molecular graph, the difference vector encodes the structural changes occurred from molecule $X$ to $Y$:

$$\delta_{X,Y}^{\mathcal{T}} = \sum_i y_i^{\mathcal{T}} - \sum_i x_i^{\mathcal{T}} \qquad \delta_{X,Y}^{\mathcal{G}} = \sum_i y_i^{\mathcal{G}} - \sum_i x_i^{\mathcal{G}} \tag{11}$$

Following Kingma & Welling (2013), the approximate posterior $Q(\cdot|X,Y)$ is modeled as a normal distribution, allowing us to sample latent codes $z^{\mathcal{T}}$ and $z^{\mathcal{G}}$ via reparameterization trick. The mean and log variance of $Q(\cdot|X,Y)$ is computed from $\delta_{X,Y}$ with two separate affine layers $\mu(\cdot)$ and $\Sigma(\cdot)$:

$$z^{\mathcal{T}} \sim \mathcal{N}\left(\mu(\delta_{X,Y}^{\mathcal{T}}), \Sigma(\delta_{X,Y}^{\mathcal{T}})\right) \qquad z^{\mathcal{G}} \sim \mathcal{N}\left(\mu(\delta_{X,Y}^{\mathcal{G}}), \Sigma(\delta_{X,Y}^{\mathcal{G}})\right) \tag{12}$$

Finally, we combine the latent code $z^{\mathcal{T}}$ and $z^{\mathcal{G}}$ with source tree and graph vectors:

$$\tilde{x}_i^{\mathcal{T}} = \tau(W_1^e x_i^{\mathcal{T}} + W_2^e z^{\mathcal{T}}) \qquad \tilde{x}_i^{\mathcal{G}} = \tau(W_3^e x_i^{\mathcal{G}} + W_4^e z^{\mathcal{G}}); \tag{13}$$

where $\tilde{x}_*^{\mathcal{T}}$ and $\tilde{x}_*^{\mathcal{G}}$ are "perturbed" tree and graph vectors of molecule $X$. The perturbed inputs are then fed into the decoder to synthesize the target molecule $\widehat{Y}$. The training objective follows a conditional variational autoencoder, including a reconstruction loss and a KL regularization term:

$$\mathcal{L}^{\text{VAE}}(X,Y) = -\mathbb{E}_{z \sim Q}[\log P(Y|z,X)] + \lambda_{\text{KL}} \mathcal{D}_{\text{KL}}[Q(z|X,Y)||P(z)] \tag{14}$$

## 4.2 ADVERSARIAL SCAFFOLD REGULARIZATION

Second, to avoid invalid translations, we force molecules decoded from latent codes $z \sim \mathcal{N}(0, I)$ to follow the distribution of the target domain through adversarial training (Goodfellow et al., 2014). The adversarial game involves two components. The discriminator tries to distinguish real molecules in the target domain from fake molecules generated by the model. The generator (i.e. our encoder-decoder) tries to generate molecules indistinguishable from the molecules in the target domain.

The main challenge is how to integrate adversarial training into our decoder, as the discrete decisions in tree and graph decoding hinder gradient propagation. To this end, we apply *adversarial regularization* over continuous representations of decoded molecular structures, derived from the hidden states in the decoder (Shen et al., 2017; Zhao et al., 2018). That is, we replace the input of the discriminator with continuous embeddings of discrete outputs. For efficiency reasons, we only enforce the adversarial regularization in the tree decoding step. As a result, the adversary only matches the scaffold structure between translated molecules and true samples.

The continuous representation is computed as follows. The decoder first predicts the label distribution $q_{root}$ of the root of tree $\widehat{\mathcal{T}}$. Starting from the root, we incrementally expand the tree, guided by topological predictions, and compute the hidden messages $\{h_{i_t,j_t}\}$ between nodes in the partial tree. At timestep $t$, the model decides to either expand a new node $j_t$ or backtrack to the parent of

---

**Algorithm 1** Adversarial Scaffold Regularization

---

1: **for** $k \leftarrow 1$ to $N$ **do** ▷ Discriminator training
2:     Sample batch $\{X^{(i)}\}_{i=1}^m \sim \mathbb{P}_\mathcal{X}$ and $\{Y^{(i)}\}_{i=1}^m \sim \mathbb{P}_\mathcal{Y}$.
3:     Let $\mathcal{T}^{(i)}$ be the junction tree of molecule $Y^{(i)}$. For each $\mathcal{T}^{(i)}$, compute its continuous representation $\boldsymbol{h}^{(i)}$ by unrolling the decoder with teacher forcing.
4:     Encode each molecule $X^{(i)}$ with latent codes $\boldsymbol{z}^{(i)} \sim \mathcal{N}(\boldsymbol{0}, \boldsymbol{I})$.
5:     For each $i$, unroll the decoder by feeding the predicted labels and tree topologies to construct the translated junction tree $\widehat{\mathcal{T}}^{(i)}$, and compute its continuous representation $\widehat{\boldsymbol{h}}^{(i)}$.
6:     Update $D(\cdot)$ by minimizing $\frac{1}{m}\sum_{i=1}^m -D(\boldsymbol{h}^{(i)}) + D(\widehat{\boldsymbol{h}}^{(i)})$ along with gradient penalty.
7: **end for**
8: Sample batch $\{X^{(i)}\}_{i=1}^m \sim \mathbb{P}_\mathcal{X}$ and $\{Y^{(i)}\}_{i=1}^m \sim \mathbb{P}_\mathcal{Y}$. ▷ Generator training
9: Repeat lines 3-5.
10: Update encoder/decoder by minimizing $\frac{1}{m}\sum_{i=1}^m D(\boldsymbol{h}^{(i)}) - D(\widehat{\boldsymbol{h}}^{(i)})$.

---

node $i_t$. We denote this binary decision as $d(i_t, j_t) = \mathbf{1}_{\boldsymbol{p}_t > 0.5}$, which is determined by the topological score $\boldsymbol{p}_t$ in Eq.(6). For the true samples $Y$, the hidden messages are computed by Eq.(3) with teacher-forcing, namely replacing the label and topological predictions with their ground truth values. For the translated samples $\widehat{Y}$ from source molecules $X$, we replace the one-hot encoding $\boldsymbol{f}_{i_t}$ with its softmax distribution $\boldsymbol{q}_{i_t}$ over cluster labels in Eq.(3) and (4). Moreover, we multiply message $\boldsymbol{h}_{i_t, j_t}$ with the binary gate $d(i_t, j_t)$, to account for the fact that the messages should depend on the topological layout of the tree:

$$\boldsymbol{h}_{i_t, j_t} = \begin{cases} d(i_t, j_t) \cdot \text{GRU}(\boldsymbol{q}_{i_t}, \{\boldsymbol{h}_{k,i_t}\}_{(k,i_t) \in \tilde{\mathcal{E}}_t, k \neq j_t}) & \text{if } j_t \text{ is a child of node } i_t \\ (1 - d(i_t, j_t)) \cdot \text{GRU}(\boldsymbol{q}_{i_t}, \{\boldsymbol{h}_{k,i_t}\}_{(k,i_t) \in \tilde{\mathcal{E}}_t, k \neq j_t}) & \text{vice versa} \end{cases} \tag{15}$$

As $d(i_t, j_t)$ is computed by a non-differentiable threshold function, we approximate its gradient with a straight-through estimator (Bengio et al., 2013; Courbariaux et al., 2016). Specifically, we replace the threshold function with a differentiable hard sigmoid function during back-propagation, while using the threshold function in the forward pass. This technique has been successfully applied to training neural networks with dynamic computational graphs (Chung et al., 2016).

Finally, after the tree $\mathcal{T}$ is completely decoded, we derive its continuous representation $\boldsymbol{h}_\mathcal{T}$ by concatenating the root label distribution $\boldsymbol{q}_{root}$ and the sum of its inward messages:

$$\boldsymbol{s}_{root} = \sum_{k \in N(root)} \boldsymbol{h}_{k,root} \qquad \boldsymbol{h}_\mathcal{T} = [\boldsymbol{q}_{root}, \boldsymbol{s}_{root}] \tag{16}$$

We implement the discriminator $D(\cdot)$ as a multi-layer feedforward network, and train the adversary using Wasserstein GAN with gradient penalty (Arjovsky et al., 2017; Gulrajani et al., 2017). The whole algorithm is described in Algorithm 1.

## 5 EXPERIMENTS

**Data** Our graph-to-graph translation models are evaluated on three molecular optimization tasks. Following standard practice in MMPA, we construct training sets by sampling molecular pairs $(X, Y)$ with significant property improvement and molecular similarity $sim(X, Y) \geq \delta$. The similarity constraint is also enforced at evaluation time to exclude arbitrary mappings that completely ignore the input $X$. We measure the molecular similarity by computing Tanimoto similarity over Morgan fingerprints (Rogers & Hahn, 2010). Next we describe how these tasks are constructed.

- **Penalized logP** We first evaluate our methods on the constrained optimization task proposed by Jin et al. (2018). The goal is to improve the penalized logP score of molecules under the similarity constraint. Following their setup, we experiment with two similarity constraints ($\delta = 0.4$ and $0.6$), and we extracted 99K and 79K translation pairs respectively from the ZINC dataset (Sterling & Irwin, 2015; Jin et al., 2018) for training. We use their validation and test sets for evaluation.
- **Drug likeness (QED)** Our second task is to improve drug likeness of compounds. Specifically, the model needs to translate molecules with QED scores (Bickerton et al., 2012) within the range

Table 1: Translation performance on penalized logP task. GCPN results are copied from You et al. (2018a). We rerun JT-VAE under our setup to ensure all results are comparable.

| Method | $\delta = 0.6$ | | $\delta = 0.4$ | |
|---|---|---|---|---|
| | Improvement | Diversity | Improvement | Diversity |
| MMPA | $1.65 \pm 1.44$ | 0.329 | $3.29 \pm 1.12$ | **0.496** |
| JT-VAE | $0.28 \pm 0.79$ | - | $1.03 \pm 1.39$ | - |
| GCPN | $0.79 \pm 0.63$ | - | $2.49 \pm 1.30$ | - |
| VSeq2Seq | $\mathbf{2.33 \pm 1.17}$ | 0.331 | $3.37 \pm 1.75$ | 0.471 |
| VJTNN | $\mathbf{2.33 \pm 1.24}$ | **0.333** | $\mathbf{3.55 \pm 1.67}$ | 0.480 |

$[0.7, 0.8]$ into the higher range $[0.9, 1.0]$. This task is challenging as the target range contains only the top 6.6% of molecules in the ZINC dataset. We extracted a training set of 88K molecule pairs with similarity constraint $\delta = 0.4$. The test set contains 800 molecules.

- **Dopamine Receptor (DRD2)** The third task is to improve a molecule's biological activity against a biological target named the dopamine type 2 receptor (DRD2). We use a trained model from Olivecrona et al. (2017) to assess the probability that a compound is active. We ask the model to translate molecules with predicted probability $p < 0.05$ into active compounds with $p > 0.5$. The active compounds represent only 1.9% of the dataset. With similarity constraint $\delta = 0.4$, we derived a training set of 34K molecular pairs from ZINC and the dataset collected by Olivecrona et al. (2017). The test set contains 1000 molecules.

**Baselines** We compare our approaches (VJTNN and VJTNN+GAN) with the following baselines:

- **MMPA**: We utilized (Dalke et al., 2018)'s implementation to perform MMPA. Molecular transformation rules are extracted from the ZINC and Olivecrona et al. (2017)'s dataset for corresponding tasks. During testing, we translate a molecule multiple times using different matching transformation rules that have the highest average property improvements in the database (Appendix B).

- **Junction Tree VAE**: Jin et al. (2018) is a state-of-the-art generative model over molecules that applies gradient ascent over the learned latent space to generate molecules with improved properties. Our encoder-decoder architecture is closely related to their autoencoder model.

- **VSeq2Seq**: Our second baseline is a variational sequence-to-sequence translation model that uses SMILES strings to represent molecules and has been successfully applied to other molecule generation tasks (Gómez-Bombarelli et al., 2016). Specifically, we augment the architecture of Bahdanau et al. (2014) with stochastic latent codes learned in the same way as our VJTNN model.

- **GCPN**: GCPN (You et al., 2018a) is a reinforcement learning based model that modifies a molecule by iteratively adding or deleting atoms and bonds. They also adopt adversarial training to enforce naturalness of the generated molecules.

**Model Configuration** Both VSeq2Seq and our models use latent codes of dimension $|\boldsymbol{z}| = 8$, and we set the KL regularization weight $\lambda_{\mathrm{KL}} = 1/|\boldsymbol{z}|$. For the VSeq2Seq model, the encoder is a one-layer bidirectional LSTM and the decoder is a one-layer LSTM with hidden state dimension 600. For fair comparison, we control the size of both VSeq2Seq and our models to be around 4M parameters. Due to limited space, we defer other hyper-parameter settings to the appendix.

### 5.1 RESULTS

We quantitatively analyze the translation accuracy, diversity, and novelty of different methods.

**Translation Accuracy** We measure the translation accuracy as follows. On the penalized logP task, we follow the same evaluation protocol as JT-VAE. That is, for each source molecule, we decode $K$ times with different latent codes $\boldsymbol{z} \sim \mathcal{N}(\boldsymbol{0}, \boldsymbol{I})$, and report the molecule having the highest property improvement under the similarity constraint. We set $K = 20$ so that it is comparable with the baselines. On the QED and DRD2 datasets, we report the success rate of the learned translations. We define a translation as successful if one of the $K$ translation candidates satisfies the similarity constraint and its property score falls in the target range (QED $\in [0.9, 1.0]$ and DRD2 $> 0.5$).

Tables 1 and 2 give the performance of all models across the three datasets. Our models outperform the MMPA baseline with a large margin across all the tasks, clearly showing the advantage

Table 2: Translation performance on QED and DRD2 task. JT-VAE and GCPN results are computed by running Jin et al. (2018) and You et al. (2018a)'s open-source implementations.

| Method | QED | | | DRD2 | | |
|--------|---------|-----------|---------|---------|-----------|---------|
| | Success | Diversity | Novelty | Success | Diversity | Novelty |
| MMPA | 32.9% | 0.236 | 99.9% | 46.4% | **0.275** | 99.9% |
| JT-VAE | 8.8% | - | - | 3.4% | - | - |
| GCPN | 9.4% | 0.216 | 100% | 4.4% | 0.152 | 100% |
| VSeq2Seq | 58.5% | 0.331 | 99.6% | 75.9% | 0.176 | 79.7% |
| VJTNN | 59.9% | 0.373 | 98.3% | 77.8% | 0.156 | 83.4% |
| VJTNN+GAN | **60.6%** | **0.376** | 99.0% | **78.4%** | 0.162 | 82.7% |

Figure 4: Examples of diverse translations learned by VJTNN+GAN on QED and DRD2 dataset.

of molecular translation approach over rule based methods. Compared to JT-VAE and GCPN baselines, our models perform significantly better because they are trained on parallel data that provides direct supervision, and therefore more sample efficient. Overall, our graph-to-graph approach performs better than the VSeq2Seq baseline, indicating the benefit of graph based representation. The proposed adversarial training method also provides slight improvement over VJTNN model. The VJTNN+GAN is only evaluated on the QED and DRD2 tasks with well-defined target domains that are explicitly constrained by property ranges.

**Diversity** We define the diversity of a set of molecules as the average pairwise Tanimoto distance between them, where Tanimoto distance $dist(X, Y) = 1 - sim(X, Y)$. For each source molecule, we translate it $K$ times (each with different latent codes), and compute the diversity over the set of validly translated molecules.[2] As we require valid translated molecules to be similar to a given compound, the diversity score is upper-bounded by the maximum allowed distance (e.g. the maximum diversity score is around 0.6 on the QED and DRD2 tasks). As shown in Tables 1 and 2, our methods achieve higher diversity score than MMPA and VSeq2Seq on two and three tasks respectively. Figure 4 shows some examples of diverse translation over the QED and DRD2 tasks.

**Novelty** Lastly, we report how often our model discovers new molecules in the target domain that are unseen during training. This is an important metric as the ultimate goal of drug discovery is to design new molecules. Let $\mathcal{M}$ be the set of molecules generated by the model and $\mathcal{S}$ be the molecules given during training. We define novelty as $1 - |\mathcal{M} \cap \mathcal{S}|/|\mathcal{S}|$. On the QED and DRD2 datasets, our models discover new compounds most of the time, but less frequently than MMPA and GCPN. Nonetheless, these methods have much lower translation success rate.

## 6 CONCLUSION

In conclusion, we have evaluated various graph-to-graph translation models for molecular optimization. By combining the variational junction tree encoder-decoder with adversarial training, we can generate better and more diverse molecules than the baselines.

---

[2]To isolate the translation accuracy from the diversity measure, we exclude the failure cases from diversity calculation, namely excluding molecules that have no valid translation. Otherwise models with lower success rates will always have lower diversity.

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

## A    MODEL ARCHITECTURE

**Tree and Graph Encoder**  For the graph encoder, functions $g_1(\cdot)$ and $g_2(\cdot)$ are parameterized as a one-layer neural network ($\tau(\cdot)$ represents the ReLU function):

$$\boldsymbol{\nu}_{uv}^{(t)} = \tau\left(\boldsymbol{W}_1^g \boldsymbol{f}_u + \boldsymbol{W}_2^g \boldsymbol{f}_{uv} + \sum_{w \in N(u)\backslash v} \boldsymbol{W}_3^g \boldsymbol{\nu}_{wu}^{(t-1)}\right) \tag{17}$$

$$\boldsymbol{x}_u = \tau\left(\boldsymbol{U}_1^g \boldsymbol{f}_u + \sum_{v \in N(u)} \boldsymbol{U}_2^g \boldsymbol{\nu}_{vu}^{(T)}\right) \tag{18}$$

For the tree encoder, since it updates the messages with more iterations, we parameterize function $g_1(\cdot)$ as a tree GRU function for learning stability (edge features $\boldsymbol{f}_{uv}$ are omitted because they are always zero). We keep the same parameterization for $g_2(\cdot)$, with a different set of parameters.

$$\boldsymbol{\nu}_{uv}^{(t)} = \text{GRU}\left(\boldsymbol{f}_u, \{\boldsymbol{\nu}_{wu}^{(t-1)}\}_{w \in N(u)\backslash v}\right) \tag{19}$$

**Tree Gated Recurrent Unit**  The tree GRU function $\text{GRU}(\cdot)$ for computing message $\boldsymbol{h}_{ij}$ in Eq.(3) is defined as follows (Jin et al., 2018):

$$\boldsymbol{s}_{ij} = \sum_{k \in N(i)\backslash j} \boldsymbol{h}_{ki} \tag{20}$$

$$\boldsymbol{z}_{ij} = \sigma(\boldsymbol{W}^z \boldsymbol{f}_i + \boldsymbol{U}^z \boldsymbol{s}_{ij} + \boldsymbol{b}^z) \tag{21}$$

$$\boldsymbol{r}_{ki} = \sigma(\boldsymbol{W}^r \boldsymbol{f}_i + \boldsymbol{U}^r \boldsymbol{h}_{ki} + \boldsymbol{b}^r) \tag{22}$$

$$\tilde{\boldsymbol{h}}_{ij} = \tanh\left(\boldsymbol{W} \boldsymbol{f}_i + \boldsymbol{U} \sum_{k \in N(i)\backslash j} \boldsymbol{r}_{ki} \odot \boldsymbol{h}_{ki} + \boldsymbol{b}\right) \tag{23}$$

$$\boldsymbol{h}_{ij} = (1 - \boldsymbol{z}_{ij}) \odot \boldsymbol{s}_{ij} + \boldsymbol{z}_{ij} \odot \tilde{\boldsymbol{h}}_{ij} \tag{24}$$

**Tree Decoder Attention**  The attention mechanism is implemented as a bilinear function between decoder state $\boldsymbol{h}_t$ and source tree and graph vectors normalized by the softmax function:

$$\boldsymbol{\alpha}_{i,t}^{\mathcal{T}} = \frac{\exp(\boldsymbol{h}_t \boldsymbol{A}_{\mathcal{T}} \boldsymbol{x}_i^{\mathcal{T}})}{\sum_k \exp(\boldsymbol{h}_t \boldsymbol{A}_{\mathcal{T}} \boldsymbol{x}_k^{\mathcal{T}})} \qquad \boldsymbol{\alpha}_{i,t}^{\mathcal{G}} = \frac{\exp(\boldsymbol{h}_t \boldsymbol{A}_{\mathcal{T}} \boldsymbol{x}_i^{\mathcal{G}})}{\sum_k \exp(\boldsymbol{h}_t \boldsymbol{A}_{\mathcal{T}} \boldsymbol{x}_k^{\mathcal{G}})} \tag{25}$$

**Graph Decoder**  We use the same graph neural architecture (Jin et al., 2018) for scoring candidate attachments. Let $G_i$ be the graph resulting from a particular merging of cluster $C_i$ in the tree with its neighbors $C_j$, $j \in N_{\widehat{\mathcal{T}}}(i)$, and let $u, v$ denote atoms in the graph $G_i$. The main challenge of attachment scoring is *local isomorphism*: Suppose there are two neighbors $C_j$ and $C_k$ with the same cluster labels. Since they share the same cluster label, exchanging the position of $C_j$ and $C_k$ will lead to isomorphic graphs. However, these two cliques are actually not exchangeable if the subtree under $j$ and $k$ are different (Illustrations can be found in Jin et al. (2018)). Therefore, we need to incorporate information about those subtrees when scoring the attachments.

To this end, we define index $\alpha_v = i$ if $v \in C_i$ and $\alpha_v = j$ if $v \in C_j \backslash C_i$. The index $\alpha_v$ is used to mark the position of the atoms in the junction tree, and to retrieve messages $\boldsymbol{h}_{i,j}$ summarizing the subtree under $i$ along the edge $(i, j)$ obtained by running the tree encoding algorithm. The tree messages are augmented into the graph message passing network to avoid local isomorphism:

$$\boldsymbol{\mu}_{uv}^{(t)} = \tau(\boldsymbol{W}_1^a \boldsymbol{f}_u + \boldsymbol{W}_2^a \boldsymbol{f}_{uv} + \boldsymbol{W}_3^a \widetilde{\boldsymbol{\mu}}_{uv}^{(t-1)}) \tag{26}$$

$$\widetilde{\boldsymbol{\mu}}_{uv}^{(t-1)} = \begin{cases} \sum_{w \in N(u)\backslash v} \boldsymbol{\mu}_{wu}^{(t-1)} & \alpha_u = \alpha_v \\ \boldsymbol{h}_{\alpha_u, \alpha_v} + \sum_{w \in N(u)\backslash v} \boldsymbol{\mu}_{wu}^{(t-1)} & \alpha_u \neq \alpha_v \end{cases} \tag{27}$$

The final representation of graph $G_i$ is $\boldsymbol{m}_{G_i} = \sum_v \boldsymbol{\mu}_v^{G_i}$, where

$$\boldsymbol{\mu}_u^{G_i} = \tau\left(\boldsymbol{U}_1^a \boldsymbol{f}_u + \sum_{v \in N(u)} \boldsymbol{U}_2^a \boldsymbol{\mu}_{vu}^{(T)}\right) \tag{28}$$

**Adversarial Scaffold Regularization**  Algorithm 2 describes the tree decoding algorithm for adversarial training. It replaces the ground truth input $\boldsymbol{f}_*$ with predicted label distributions $\boldsymbol{q}_*$, enabling gradient propagation from the discriminator.

---

**Algorithm 2** Soft Tree Decoding for Adversarial Regularization

---

**Require:** Source tree and graph vectors $\{\boldsymbol{x}_*^{\mathcal{T}}\}, \{\boldsymbol{x}_*^{\mathcal{G}}\}$
 1: **Initialize:** Tree $\widehat{\mathcal{T}} \leftarrow \emptyset$; Global counter $t \leftarrow 0$
 2: **function** DecodeTree($i$)
 3:    **repeat**
 4:       $t \leftarrow t + 1$
 5:       Predict topology score $\boldsymbol{p}_t$ with Eq.(6), replacing $\boldsymbol{f}_i$ with predicted label distribution $\boldsymbol{q}_i$.
 6:       **if** $\boldsymbol{p}_t \geq 0.5$ **then**
 7:          Create a child $j$ and add it to tree $\widehat{\mathcal{T}}$.
 8:          Predict the node label distribution $\boldsymbol{q}_j$ with Eq.(9)
 9:          Compute message $\boldsymbol{h}_{i,j}$ with Eq.(15)
10:          DecodeTree($j$)
11:       **end if**
12:    **until** $\boldsymbol{p}_t < 0.5$
13:    Let $j$ be the parent node of $i$. Compute message $\boldsymbol{h}_{i,j}$ with Eq.(15)
14: **end function**

---

# B EXPERIMENTAL DETAILS

**Training Details** We elaborate on the hyper-parameters used in our experiments. For our models, the hidden state dimension is 300 and latent code dimension $|\boldsymbol{z}| = 8$. The tree encoder runs message passing for 6 iterations, and graph encoder runs for 3 iterations. The entire model has 3.9M parameters. For VSeq2Seq, the encoder is a one-layer bidirectional LSTM and the decoder is a one-layer uni-directional LSTM. The attention scores are computed in the same way as Bahdanau et al. (2014). We set the hidden state dimension of the recurrent encoder and decoder to be 600, with 4.2M parameters in total.

All models are trained with the Adam optimizer for 20 epochs with learning rate 0.001. We anneal the learning rate by 0.9 for every epoch. For adversarial training, our discriminator is a three-layer feed-forward network with hidden layer dimension 300 and LeakyReLU activation function. The discriminator is trained for $N = 5$ iterations with gradient penalty weight $\beta = 10$.

**Property Calculation** The penalized logP is calculated using You et al. (2018a)'s implementation, which utilizes RDKit (Landrum, 2006) to compute clogP and synthetic accessibility scores. The QED scores are also computed using RDKit's built-in functionality. The DRD2 activity prediction model is downloaded from `https://github.com/MarcusOlivecrona/REINVENT/blob/master/data/clf.pkl`.

**MMPA Procedure** We utilized the open source toolkit `mmpdb` (Dalke et al., 2018) to perform matching molecular pair (MMP) analysis (`https://github.com/rdkit/mmpdb`). On the logP and QED tasks, we constructed a database of transformation rules extracted from the ZINC dataset (with test set molecules excluded). On the DRD2 task, the database is constructed from both ZINC and the dataset from Olivecrona et al. (2017). During testing, each molecule is translated $K = 20$ times with different matching rules. When there are more than $K$ matching rules, we choose those with the highest average property improvement. This statistic is calculated during database construction.

**Dataset Curation** The training set of the penalized logP task is curated from the ZINC dataset of 250K molecules (Jin et al., 2018). A molecular pair $(X, Y)$ is selected into the training set if the Tanimoto similarity $sim(X, Y) \geq \delta$ and the property improvement is significant enough (greater than certain threshold). On the QED and DRD2 tasks, we select training molecular pairs $(X, Y)$ if $sim(X, Y) \geq 0.4$ and both $X$ and $Y$ fall into the source and target property range. For each task, we ensured that all molecules in validation and test set had never appeared during training.

