# OpenReview forum: "Learning Multimodal Graph-to-Graph Translation for Molecule Optimization"
_ICLR.cc/2019/Conference_

### Official Review · AnonReviewer1 · 2018-11-02
**A paper proposing a quite complex system (with no explicit probabilistic factorisation) which seems to obtain good experimental results.**

**Rating:** 6
**Confidence:** 4

**Review:**

As a reviewer I am expert in learning in structured data domains.
The paper proposes a quite complex system, involving many different choices and components, for obtaining chemical compounds with improved properties starting from a given corpora.
Overall presentation is good, although some details/explanations/motivations are missing. I guess this was due to the need to keep the description of a quite complex system in the given space limit. Such details/explanations/motivations could, however, have been inserted in the appendix. As an example, let consider the description of the decoding of the junction tree. In that section, it is not explained when the decoding process stops. My understanding is that this is when, being in the root node, the choice is to go back to the parent (that does not exist). In the same section, it is not explicitly discussed that the probability to select between adding a node or going back to the parent should have a different distribution according to "how many" nodes have been generated before, i.e. we do not want to have a high probability to "go back" at the beginning of the decoding, while I guess it is desirable that such probability increases proportionally with the number of generated nodes. This leads to an issue that I personally think is important: the paper does lack an explicit probabilistic modelling of the different involved components, which may help for a better understanding of all the assumptions made in the construction of the proposed system.
The complexity of the proposed system is actually an issue since the author(s) do not attempt (except for  the presence or absence of the adversarial scaffold regularization and the number of trials in appendix) an analysis of the influence of the different components (and corresponding hyper-parameters).
Reference to previous relevant work seems to be complete.
I think the paper is relevant for ICLR (although there is no explicit analysis of the obtained hidden representations) and of interest for a good portion of attendees.

Minor issues:
- Tree and Graph Encoding: asynchronous update implies that T should be a multiple of the diameter of the input graph to guarantee a proper propagation of information across the graph. A discussion about that would be needed.
- eq.(6): \mathbb{u}^d is not defined.
- Section 3.3:
   - first paragraph is not clear. An example and/or figure is needed to understand the argument, which is related to the presence of cycles.
  - the definition of f(G_i) involves  \mathbb{x}_u. I guess they should be  \mathbb{x}_u^G.
  - not clear how the log-likelihood of ground truth subgraphs is computed given that the predicted junction tree, especially at the beginning of training, may be way different from the correct one. Moreover, what is the assumed bias of this choice ?
- Table I: please provide an explanation of why using a larger value for \delta does provide worst performance than a smaller value. From an optimisation point of view it should provide at least an as good performance. This is a clear indication that the used procedure is suboptimal.
- diversity could be influenced by the cardinality of the sample. Is this false ? please discuss why diversity is (not) biased versus larger sets.

---

> ### Author Response · Authors · 2018-11-13
> **Response to Reviewer 1 (Part II): Clarifications with paper updated to elaborate Section 3.3**
>
> Thank you very much for your insightful comments. Our response to the issues you mentioned is the following:
>
> 1) Please provide an explanation of why using a larger value for delta gives worse performance than a smaller value.
> A larger delta implies a tighter similarity constraint. For instance, setting delta to 0.6 means the generated compounds Y have to be very similar to the input molecule X (sim(X,Y) > 0.6). When delta decreases to 0.4, the generated structures are allowed to deviate more from the starting point X (sim(X,Y) > 0.4). Therefore, one would naturally expect the model to perform better (find higher scoring molecules) when delta is smaller since the structures can be chosen from a larger set.
>
> 2) Diversity could be influenced by the cardinality of the sample. Please discuss why diversity is (not) biased versus larger sets.
> We agree that the diversity depends on the sample size. Therefore, all the models are evaluated with the same sample size (K=50) for fair comparison. That is, for each molecule in the test set, we randomly sample 50 times from each model to compute the resulting diversity score.
>
> 3) Tree and graph encoding: asynchronous update implies that T should be a multiple of the diameter of the input graph to guarantee a proper propagation of information across the graph.
> We agree that a number of iterations (T) is required for proper propagation of information across the input graph. However, T does not need to be larger than the diameter since we adopted an attention mechanism in the decoder. It can dynamically read the information across the input graph in different decoding steps. In fact, a large T (e.g., the diameter) may potentially lead to overfitting.
>
> 4) Clarification of tree decoding step (Section 3.2)
> First, the tree decoding process stops when it choose to backtrack at the root node. Second, we agree that this probability should depend on the number of nodes having been generated. This is implicitly captured by the neural message passing procedure. As noted in Eq. (4), the model makes this decision (expanding a new node or not) based on all the incoming messages at the current node. The messages carry information about the current (partial) tree structure, including potentially the number of nodes generated so far though not explicitly.
>
> 5) Explanation of graph decoding step (Section 3.3)
> We added Figure 2 to illustrate why the graph decoding step is not deterministic and how one junction tree can be decoded into different molecular graphs. Regarding the likelihood of ground truth subgraphs, we applied teacher forcing, i.e., we feed the graph decoder with ground truth junction trees as input. Section 3.3 has been updated correspondingly.

---

> ### Author Response · Authors · 2018-11-13
> **Response to Reviewer 1 (Part I): Probabilistic modeling of the involved components**
>
> Thank you very much for your insightful comments. We want to provide more explanations on the probabilistic modeling of different involved components.
>
> 1) Explicit probabilistic modeling of junction tree encoder-decoder (Section 3).
> Prior work (Jin et al. 2018) found that it is beneficial to adopt a coarse-to-fine approach to generate molecular graphs: first generate the backbone structure (i.e., junction tree T) and then assemble the sub-graphs in the tree into a complete molecular graph Y. Thus
>                                                   p(Y | X) =  \sum_T p(Y | T, X) p(T | X)
> where p(Y | T, X) is the graph decoder and p(T | X) is the tree decoder. As the junction tree T of any graph is constructed through a deterministic tree decomposition algorithm, T does not function as a latent variable during training but is rather an intermediate object that can be predicted via supervised learning. Therefore,
>                                               p(Y | X) \approx p(Y | T_y, X) * p(T_y | X)
> where T_y is the junction tree underlying the target graph Y.
>
> The tree decoder generates a tree in an autoregressive manner, based on a specific sequentialization of the tree structure. A tree T is laid out as a sequence of edges {(i_1, j_1), …, (i_m, j_m)} visited in the depth-first traversal over the tree. The probability of generating T is thus
>                           p(T | X) = \prod_t  p( (i_t, j_t) | (i_1, j_1), …, (i_t-1, j_t-1), X )
> where j_t always equals i_{t+1}. Probability of (i_t, j_t) depends on two factors: 1) whether j_t is a new node; 2) If j_t is a new node, what is its label; These two factors are modeled by the topological predictor (Eq. 4-6) and the label predictor (Eq. 8-9). The message passing procedure (Eq. 3) embeds the current partial tree realized by {(i_1, j_1), …, (i_t-1, j_t-1)} into a continuous representation. Beyond the above architecture, in this paper we introduced an attention mechanism to capture how the decoded tree unravels step-by-step in an input graph X dependent manner.
>
> The graph decoder models the conditional probability p(Y | T_y, X). This is a structured prediction task since Y is a graph. The variables in this structured prediction problem are node assembling decisions between neighboring nodes in the tree. For efficiency reasons, the assembling decisions are solved locally, starting from the root and its direct neighbors. In other words, p(Y | T_y, X) is a product of probabilities of choosing the right graph attachments with each node’s neighbors, resulting in Eq. (10) (after taking log).
>
> 2) Probabilistic modeling of multi-modal translation model (Section 4)
> In this paper, we aim to learn diverse multi-modal mappings between two molecular domains, as there are many different ways to improve a given molecule. This diversity is introduced via latent variables z:
>                                            p(Y | X) = \int_z p(Y | X, z) p(z) dz
> where prior p(z) models diverse strategies of improvement, independent of X, and is taken to be a standard Gaussian distribution. The overall model resembles a conditional variational autoencoder, learnable through reparameterization (Section 4.1). The approximate posterior Q(z | Y) only depends on the target Y so as to force z to capture resulting type of molecule, inferable from Y alone.
>
> The proposed adversarial training technique (Section 4.2) is an additional regularization trying to discourage the model from generating undesirable outputs (e.g. molecules outside of the defined target domain). As a side note, p(Y | X, z) can be expanded as
>                                        p(Y | X, z) = p(Y | T_y, X, z) p(T_y | X, z)
> where latent variable z is concatenated with the encoded representation of X (Eq. 11).

---

### Official Review · AnonReviewer3 · 2018-11-02
**Interesting idea, issues in the execution**

**Rating:** 7
**Confidence:** 5

**Review:**

Update:
The score has been updated to reflect the authors' great efforts in improving the manuscript. This reviewer would suggest to accept the paper now.


Old Review Below:

The paper describes a graph-to-graph translation model for molecule optimization inspired from matched molecular pair analysis, which is an established approach for optimizing the properties of molecules. The model extends a chemistry-specific variational autoencoder architecture, and is assessed on a set of three benchmark tasks.


While the idea of manuscript is interesting and promising for bioinformatics, there are several outstanding problems, which have to be addressed before it can be considered to be an acceptable submission. This referee is willing to adjust their rating if the raised points are addressed. Overall, the paper might also be more suited at a domain-specific bioinformatics conference.


Most importantly, the paper makes several claims that are currently not backed up by experiments and/or data.

First, the authors claim that MMPs “only covers the most simple and common transformation patterns”. This is not correct, since these MMP patterns can be as complex as desired. Also, it is claimed that the presented model is able to “learn far more complex transformations than hard-coded rules”. The authors will need to provide compelling evidence to back up these claims. At least, a comparison with a traditional MMPA method needs to be performed, and added as a baseline. Also, it has to be kept in mind that the reason MMPA was introduced was to provide an easily interpretable method, which performs only local transformations at one part of the molecule. “Far more complex transformations” may thus not be desirable in the context of MMPA. Can the authors comment on that?

Second, the authors state that they “sidestep” the problem of non-generalizing property predictors in reinforcement learning, by “unifying graph generation and property estimation in one model”. How does the authors’ model not suffer from the same problem? Can they provide evidence that their model is better in property estimation than other models?


In the first benchmark (logP) the GCPN baseline is shown, but in the second benchmark table, the GCPN baseline is missing. Why? The GCPN baseline will need to be added there. Can the authors also comment on how they ensure the comparison to the GPCN and VSeq2Seq is fair? Also, can the authors comment on why they think the penalized logP task is a good benchmark?

Also, the authors write that Jin et al ICML 2018 (JTVAE) is a state of the model. However, also Liu et al NIPS 2018 (CGVAE) state that their model is state of the art. Unfortunately, both JTVAE and CGVAE were never compared against the strongest literature method so far, by Popova et al, which was evaluated on a much more challenging set of tasks than JT-VAE and CGVAE. The authors cite this paper but do not compare against it, which should to be rectified. This referee understands it is more compelling to invent new models, but currently, the literature of generative models for molecules is in a state of anarchy due to lack of solid comparison studies, which is not doing the community a great service.


Furthermore, the training details are not described in enough detail.
How exactly are the pairs selected? Where do the properties for the molecules come from? Were they calculated using the logP, QED and DRD2 models? How many molecules are used in total in each of these tasks?

---

> ### Author Response · Authors · 2018-11-11
> **Response to Reviewer 3 (Part II): Response to your other comments and questions**
>
> Thank you very much for your insightful comments. Regarding your other comments and questions, our response is the following:
>
> 1) “The authors claim that MMPs “only covers the most simple and common transformation patterns”. This is not correct, since these MMP patterns can be as complex as desired.”
> We agree that MMP patterns can be as complex as desired. However, allowing the patterns to be arbitrarily complex will result in a huge number of transformation rules. For instance, we have extracted 12 million rules in total on the logP and QED tasks when no constraints are imposed. Therefore, we have updated this claim in the paper with the following statement: “MMPA's main drawback is that large numbers of rules have to be realized (e.g. millions) to cover all the complex transformation patterns.”
>
> 2) “the reason MMPA was introduced was to provide an easily interpretable method, which performs only local transformations at one part of the molecule. ‘Far more complex transformations’ may thus not be desirable in the context of MMPA.”
> Yes, we agree that there is always a trade-off between simple and understandable rules vs performance, and that the same trade-off is present in other machine learning applications (e.g., shallow decision trees vs  neural networks). Our focus in this paper is on demonstrating the performance gains we can obtain by reformulating the task as a translation problem. Deriving interpretable explanations for the predictions is clearly an important future direction, but is orthogonal to our current effort.
>
> 3) “The authors state that they “sidestep” the problem of non-generalizing property predictors in reinforcement learning …  How does the authors’ model not suffer from the same problem? Can they provide evidence that their model is better in property estimation than other models?”
> We want to clarify that our model does not explicitly estimate the properties. As a result, we can only provide indirect evidence showing that our model can nevertheless outperform other models in mapping precursor molecules into the target set of molecules with better properties.
>
> 4) “Can the authors also comment on how they ensure the comparison to the GCPN and VSeq2Seq is fair?”
> When comparing to VSeq2Seq, we ensure that all models have about the same number of parameters (3.8~3.9 million), trained on the same dataset with the same optimizer and the same number of epochs. Both models are evaluated with K=50 translation attempts for each test compound.
> Regarding GCPN, their exact setup is not provided. As described in their paper [4], GCPN was trained in an environment whose initial state is one of the test set molecule of the logP task. They kept all the molecules generated during training and reported the molecule with the best logP improvement. We think this may bring more advantage to GCPN in our comparison, as our models do not have access to the test set.
>
> 5) “Can the authors comment on why they think the penalized logP task is a good benchmark?”
> We evaluated on this task because some prior work (e.g. JT-VAE, GCPN) has been tested on this benchmark, and their results are readily available for comparison. Indeed, this benchmark itself is not comprehensive enough. We therefore tested on two more tasks (QED and DRD2) aiming to provide a more thorough evaluation.
>
> 6) “How exactly are the pairs selected? Where do the properties for the molecules come from? Were they calculated using the logP, QED and DRD2 models? How many molecules are used …?”
> Those details have been discussed in the Appendix B. We updated the relevant paragraphs to make it more clear. To summarize, logP and QED scores are calculated with RDKit built-in functions. For DRD2 activity prediction, we directly used the pre-trained model in Olivecrona et al. [3].
> On the QED and DRD2 tasks, a molecular pair (X,Y) is selected if the Tanimoto similarity sim(X,Y) >= 0.4 and both X and Y fall into the source and target property range. On the logP task, we select molecular pairs when similarity sim(X,Y) >= delta and property improvement is greater than 0.5 (if delta=0.6) and 2.5 (if delta=0.4). In total 250K molecules are used for constructing the training pairs in the logP and QED tasks, and 350K molecules in the DRD2 task.
>
> References
> [1]  A. Dalke, J. Hert, C. Kramer. mmpdb: An Open-Source Matched Molecular Pair Platform for Large Multiproperty Data Sets. J. Chem. Inf. Model., 2018, 58 (5), pp 902–910.
> [2]  M. Popova, O. Isayev, and A. Tropsha. Deep reinforcement learning for de novo drug design. Science advances, 4(7):eaap7885, 2018.
> [3]  M. Olivecrona, T. Blaschke, O. Engkvist, and H. Chen. Molecular de-novo design through deep reinforcement learning. Journal of cheminformatics, 9(1):48, 2017.
> [4]  J. You, B. Liu, R. Ying, V. Pande, and J. Leskovec. Graph convolutional policy network for goal-directed molecular graph generation. arXiv preprint arXiv:1806.02473, 2018

---

> > ### Comment · AnonReviewer3 · 2018-11-25
> > **reply**
> >
> > First, thanks a lot for the authors efforts, this is much appreciated!
> > Nevertheless, this reviewer thinks the paper is still overselling the results, and hides limitations, which is unfortunate and unnecessary, since the modeling idea is actually promising.
> >
> >
> > Comments:
> >
> > In terms of modeling, there is indeed a distinction between mapping from molecules to better molecules over other generative models, e.g. variational autoencoders or graph-convolutional policy networks.
> >
> > However, in practice, there is no distinction, since *in effect* both models perform the optimization of molecular properties with respect to the molecules. In fact, the same scoring functions that are used in this paper here could be used by a VAE+Bayesian optimization or an RL model as the reward, and are applied in practice to hit/lead optimization as well as library generation. The former application is even more frequent in practice than the latter.
> >
> > Comments to the authors comments:
> >
> > Re: I 1)
> > Thanks for running the mmpdb baseline! A few questions on that:
> >
> > a) How did the authors optimize the hyperparameters of the mmpdb algorithm?
> > b) This reviewer does not fully understand why the authors need to translate each molecule 50 times? MMPA is determistic, so one should just need to translate once and then pick the top 50 translated/transformed molecules with the highest expected improvement that are within the similarity constraint. Can the authors comment on that in more detail?
> >
> >
> > Re I 2: Thank you for running the GCPN baseline!
> > Please note that Popova’s work, GCPN or any other comparable RL framework can be applied in straightforward way to lead optimization as well: One would just plugin a reward function of f(mol) = min( sim(startmol, mol),threshold ) + Property(mol), and wouldn’t actually have to worry about pretraining, Wouldn’t this be even more flexible & general compared to the method presented here?
> >
> > Re II 2: This reviewer remains unconvinced. This paragraph needs to be fixed in the manuscript because also implicit estimation is estimation.
> >
> > Re II 6:
> > So, if this reviewer understands correctly, the authors have scored all 250k/350k molecules using logP/QED/the DRD SVM, which are exactly the “suboptimal property predictors” that BayesOpt/RL would use for scoring, and then created pairs from them? Doesn’t this imply the same suboptimal estimation is now baked into the translation model, but implicitly?
> >
> >
> > Also in the (commendable) ablation study in the appendix, the authors state that “In a real-world drug discovery setting, there is usually a budget on how many drug candidates can be tested in the laboratory, as biological experiments are time-consuming in general. […] This is beneficial as it requires fewer experiments in the real scenario.”, but then require 250k/350k samples to train the model. Isn’t this a contradiction?
> >
> >
> > Overall:
> > So to be crystal-clear: The authors will need to remove any claims to practical drug discovery, and position their paper more realistically, then this reviewer will recommend acceptance. But in the current form, there are still too many unsupported and misleading claims.

---

> > > ### Author Response · Authors · 2018-11-26
> > > **Paper updated based on your feedbacks**
> > >
> > > Thank you very much for your insightful comments. We have removed claims about practical drug discovery, as well as claims that are not well supported by our current manuscript. For instance, we have modified the related work section (see point 3) and removed statements in the ablation study paragraph in the appendix (see point 5). We also updated statements in the experiment section since we have added MMPA and GCPN baselines.
> > >
> > > 1a) How did the authors optimize the hyperparameters of the mmpdb algorithm?
> > > The current mmpdb program is very expensive to run. It takes about 4-5 hours to perform MMPA on 1000 molecules due to large number of extracted rules. Therefore, we performed limited amount of hyperparameter tuning on the validation set to find good hyperparameters. Moreover, some hyperparameters (e.g., the size of environment fingerprints) are hard-coded in the source code, and we couldn’t investigate how these hyperparameters will affect the model performance.
> > >
> > > 1b) Why the authors need to translate each molecule 50 times? MMPA is deterministic, so one should just need to translate once and then pick the top 50 translated/transformed molecules with the highest expected improvement ...
> > > We did exactly what you describe here. Each test set molecule is translated “once”, but in this “one-time” translation, multiple matching transformation rules are applied to this compound. And we simply picked the top 50 transformed molecules within the similarity constraint. We defined “one” translation in MMPA as applying “one” transformation rule.
> > >
> > > 2) Popova’s work, GCPN or any other comparable RL framework can be applied in straightforward way to lead optimization as well: One would just plugin a reward function of f(mol) = min( sim(startmol, mol),threshold ) + Property(mol) [...] Wouldn’t this be even more flexible & general compared to the method presented here?
> > > We agree that RL framework could be extended to our conditional translation scenario. However, adding similarity into the reward itself is not enough, unless you also feed the “startmol” into the RL model so that it knows what the starting molecule looks like. Otherwise the RL model will get confused since the reward function will keep changing as the starting molecule changes during training. Therefore a successful extension of this algorithm would be a contribution in its own right.
> > >
> > > 3) Re II 2: This reviewer remains unconvinced. This paragraph needs to be fixed in the manuscript because also implicit estimation is estimation.
> > > We suppose that you are referring to our response Part II 3 (not II 2, which is about MMPA instead of implicit property estimation). We agree that current manuscript does not provide enough evidence regarding this point. Therefore we have changed the paragraph in related work section. We removed statements involving “suboptimal property estimator”.
> > >
> > > 4) The authors have scored all 250k/350k molecules using logP/QED/the DRD SVM, which are exactly the “suboptimal property predictors” that BayesOpt/RL would use for scoring, and then created pairs from them? Doesn’t this imply the same suboptimal estimation is now baked into the translation model, but implicitly?
> > > We agree that the suboptimal property estimator can implicitly affect our model, given the way we created the training data. Therefore, we have removed these claims (see point 3). However, our graph-to-graph translation model can be trained on molecular pairs constructed based on their measured properties without any property estimation models. We couldn’t do this experiment as such datasets are not publicly available, but they often exist in pharma companies. In contrast, prior models require property predictor to be an integral part of the model.
> > >
> > > 5) The authors state that “In a real-world drug discovery setting, there is usually a budget on how many drug candidates can be tested in the laboratory […] This is beneficial as it requires fewer experiments in the real scenario.”, but then require 250k/350k samples to train the model. Isn’t this a contradiction?
> > > We have removed these sentences as they can be misleading and they are irrelevant to the ablation comparison. Please note that the goal of this ablation study is to investigate the importance of the adversarial learning component.
> > > Regarding your “contradiction” concern, we used 250k/350k samples as they were readily available. The question of data efficiency applies to all neural models, including RL models for drug discovery and neural models for property prediction. To investigate this, we trained VJTNN on the logP task (delta=0.4) using only 3k molecular pairs, as compared to 120k pairs extracted from the full dataset. The test set result is 1.26 +/- 1.53 (full dataset performance was 3.3 +/- 1.8). Indeed, learning graph translation is challenging under low-resource scenario, and we leave this issue for future work.

---

> > > > ### Comment · AnonReviewer3 · 2018-11-26
> > > > **score updated**
> > > >
> > > > Thank you for updating the paper. I've updated the score as well.

---

> > > > > ### Author Response · Authors · 2018-11-28
> > > > > **Thank you!**
> > > > >
> > > > > Thank you for your insightful comments again! They are very helpful!

---

> ### Author Response · Authors · 2018-11-11
> **Response to Reviewer 3 (Part I): Required experiments added and paper updated**
>
> Thank you very much for your insightful comments. We’d like to clarify first that our model is a conditional graph-to-graph translation model which maps a given precursor compound to another with more desirable properties. Our translation approach is therefore NOT equivalent to a generative model over molecular structures (i.e., for chemical library design). This conditional translation model is useful and important for hit/lead compound optimization.
>
> In response to your suggestions, we added two additional experiments:
> 1) MMPA baseline: We utilized the open source tool “mmpdb” [1] to perform MMPA. For each task, we constructed a database of transformation rules extracted from the ZINC and Olivecrona et al. [3]’s dataset. Same as our methods, each test set molecule is translated 50 times using the matching rules found in the database. When there are more than 50 matching rules, we choose those having higher average property improvement in the database. This statistic is calculated during the database construction. More details can be found in the Appendix B.
>
> The results are shown in Tables 1 and 2 in the updated paper. On the QED and DRD2 tasks, our model outperforms MMPA with significant margin in terms of translation success rate (56.9% vs 20.8% on QED and 81.0% vs 35.6% on DRD2). On the logP task, our model also outperforms MMPA in terms of average property improvement (3.37 vs 2.00 when delta=0.4 and 1.53 vs 1.41 when delta=0.6).
>
> 2) GCPN baseline: We used You et al [4]’s open source implementation to train GCPN on the QED and DRD2 tasks. As stated in their paper [4], GCPN was trained in an environment whose initial state is one of the test set molecules. They kept all the molecules generated during training and reported the molecule with the best property improvement. For consistency, we adopted the same strategy in training and evaluation of GCPN (i.e., training on the test set of QED and DRD2). The performance is reported in Table 2. Our model greatly outperforms GCPN (56.9% vs 9.4% on QED and 81.0% vs 4.4% on DRD2).
>
> Regarding Popova et al.’s method [2], we have carefully read the paper and studied its open-sourced code. The model described in [2] is not directly applicable to our setting as it targets chemical library design while our focus is on lead optimization starting from a given precursor compound. Their model architecture would have to be modified so as to take a precursor compound as an input to be optimized / translated. In fact, Popova et al. list this task as a future work.
>
> Due to limited length, our response to your other questions is posted in another post.
>
> References
> [1]  A. Dalke, J. Hert, C. Kramer. mmpdb: An Open-Source Matched Molecular Pair Platform for Large Multiproperty Data Sets. J. Chem. Inf. Model., 2018, 58 (5), pp 902–910.
> [2]  M. Popova, O. Isayev, and A. Tropsha. Deep reinforcement learning for de novo drug design. Science advances, 4(7):eaap7885, 2018.
> [3]  M. Olivecrona, T. Blaschke, O. Engkvist, and H. Chen. Molecular de-novo design through deep reinforcement learning. Journal of cheminformatics, 9(1):48, 2017.
> [4]  J. You, B. Liu, R. Ying, V. Pande, and J. Leskovec. Graph convolutional policy network for goal-directed molecular graph generation. arXiv preprint arXiv:1806.02473, 2018

---

### Official Review · AnonReviewer2 · 2018-11-03
**review on "Learning Multimodal Graph-to-Graph Translation for Molecule Optimization"**

**Rating:** 7
**Confidence:** 4

**Review:**

This paper proposed an extension of JT-VAE [1] into the graph to graph translation scenario. To help make the translation model predicting diverse and valid outcomes, the author added the latent variable to capture the multi-modality, and an adversarial regularization in the latent space. Experiment on molecule translation tasks show significant improvement over existing methods.

The paper is well written. The author explains the GNN, JT-VAE and GAN in a very organized way. The idea of modeling the molecule optimization as translation problem is interesting, and sounds more promising (and could be easier) than finding promising molecule from scratch.

Technically I think it is reasonable to use latent variable model to handle the multi-modality. Using GAN to align the distribution is also a well adapted method recently. Thus overall the method is not too surprising to me, but the paper executes it nicely. Given the significant empirical improvement, I think this paper has made a valid contribution to the area.

Regarding the results in Table 1, I’m curious why the VSeq2Seq is better than JT-VAE and GCPN (given the latter two are the current state-of-the-art)?

Another thing I’m curious about is the ‘stacking’ of this translation model. Suppose we keep translating the molecule X1 -> X2 -> X3 ...  using the learned translation model, would the model still gets improvement after X2? When would it get maxed out?
Or if we train with ‘path’ translation (i.e., train with improvement path with variable length), instead of just the pair translation, would that be helpful? I’m not asking for more experiments, but some discussion might be useful.

[1] Jin et.al, Junction tree variational autoencoder for molecular graph generation, ICML 2018

---

> ### Author Response · Authors · 2018-11-13
> **Response to Reviewer 2: Explanation to your questions**
>
> Thank you very much for your insightful comments.
>
> 1) Why VSeq2Seq is better than JT-VAE and GCPN?
> The main reason is that VSeq2Seq is trained with direct translation pairs through supervised learning, while JT-VAE and GCPN have to learn to discover these pairs in a weakly supervised manner. For instance, GCPN iteratively modifies a given molecule to maximize the predicted property score, where the translation pairs are discovered through reinforcement learning. JT-VAE optimizes a molecule by first mapping it into its latent representation and then performing gradient ascent in the latent space. In this case, translation pairs are discovered through the gradient signal given by the property predictor, which is trained on molecules with labeled properties. As the models are evaluated by translation quality, training the model directly with translation pairs is advantageous.
>
> 2) Suppose we keep translating the molecule X1 -> X2 -> X3 ...  using the learned translation model, would the model still get improvement after X2? When would it get maxed out?
> On the logP task, the model may still get improvements after X2, but we suspect this process will get maxed out after several steps because in general it is harder to optimize a molecule with high property scores. The QED and DRD2 tasks are different from logP task, as the target domain now becomes a closed set defined by the property range. As long as X2 belongs to the target domain (e.g., QED >= 0.9, DRD2 >= 0.5), this process will get maxed out since the model is trained only to improve molecules outside of the target domain.
>
> 3) If we train with ‘path’ translation (i.e., train with improvement path with variable length), instead of just the pair translation, would that be helpful?
> In general, it is harder to collect ‘path’ translation data than translation pairs due to data sparsity. For instance, to find a translation path X1 -> X2 -> X3, we need (X1,X2) and (X2,X3) to be valid translation pairs (i.e., both pairs satisfying property improvement and similarity constraints). Nonetheless, we believe that training the model with path translation will be helpful for global optimization -- finding molecules with the best property scores in the entire molecular space.

---

### Meta-Review · Area_Chair1 · 2018-12-13
**after revisions the reviewers reached a consensus on accepting the paper**

**Confidence:** 5
**Recommendation:** Accept (Poster)

**Metareview:**

The revisions made by the authors convinced the reviewers to all recommend accepting this paper. Therefore, I am recommending acceptance as well. I believe the revisions were important to make since I concur with several points in the initial reviews about additional baselines. It is all too easy to add confusion to the literature by not including enough experiments.